# Conceptual Model, Experiment and Numerical Simulation of Diaphragm Wall Leakage Detection Using Distributed Optical Fiber

**DOI:** 10.3390/ma16020561

**Published:** 2023-01-06

**Authors:** Jianxiu Wang, Pengfei Liu, Rui Xue, Weiqiang Pan, Ansheng Cao, Yanxia Long, Huboqiang Li, Yuanwei Sun

**Affiliations:** 1College of Civil Engineering, Tongji University, Shanghai 200092, China; 2Key Laboratory of Geotechnical and Underground Engineering of Ministry of Education, Tongji University, Shanghai 200092, China; 3Shanghai Tunnel Engineering Company Co., Ltd., Shanghai 200082, China

**Keywords:** foundation pit, leakage, distributed optical fiber, thermal leakage detection, experiment, numerical simulation

## Abstract

Leakage in the diaphragm wall is difficult to detect in deep foundation pits. In this study, the conceptual model of active and passive thermal leak detection methods was proposed according to the occurrence of temperature field anomalies caused by seepage. Experiments were performed using a heating system and an optical fiber temperature measurement system to verify the thermal leakage detection systems. Numerical simulations were performed to understand the mechanism of the detecting method. Results indicated that the optical cable could detect the low-temperature anomaly in the active temperature field leak detection. The arrangement method of the leakage detection system was also presented in actual engineering.

## 1. Introduction

Diaphragm walls are widely used in water-rich soft strata for urban deep foundation pit engineering because they are waterproof and have excellent retaining ability. However, the joint of the wall can easily produce construction quality defects because of the characteristics of the construction technology of the diaphragm wall [1,2]. Moreover, wall joint leakage, water inrush, quicksand, and other engineering risks appear after the excavation of the foundation pit. Seepage of underground engineering is an important safety problem, which may cause internal erosion of the structure. Therefore, leakage detection for a diaphragm wall is performed to ensure the safety and stability of surrounding buildings and the safety of underground engineering space development and construction. The leakage of the diaphragm wall used as a waterproof curtain must be detected before the excavation of foundation pit engineering [3,4,5].

Given the urgency required in construction practice, the leakage detection technology for diaphragm wall joints is being developed constantly. At present, the detection methods include leakage monitoring based on capacitive sensors [6], electromagnetic leakage detection technology [7], tracer method leak detection technology [8], and distributed fiber optic percolation detection technology [9]. Du [10] monitored the infiltration process of liquid in rock and soil by using the direct current resistivity method. The regional scope of liquid seepage at each time was determined. Moreover, the priority path of liquid seepage at each time and the change in water in each region were determined. This method mainly aims at point source infiltration. The actual water infiltration is a complex process. Cai et al. [11] took soil resistivity as the main research variable. They used a feasible micro-voltage method to verify the foundation pit leakage model test. Wan et al. [12] applied the electromagnetic resistivity method in detecting potential leakage hazards in the dam where electrodes were laid upstream and downstream of the dyke. They formed a current loop using a leakage path by constructing a dam leakage detection model. A three-component spatial distribution of magnetic anomaly with different leakage locations and different leakage radii was obtained. The disadvantage of the resistivity method is that the accuracy is low, and multiple repeated measurements are required to ensure the reliability of the detection. The real-time performance is poor, and the accuracy decreases with the increase of the monitoring depth [13,14,15]. Tomkovic et al. [16] used the isotope tracer method to measure seepage velocity and flow direction. However, it cannot be popularized in engineering construction and operation because of the influence of radioactive isotopes on the environment and the human body. Dorjsuren et al. [17] adopted the temperature gradient method to obtain an accurate river flow estimation. However, it is only effective in strong seepage formation. Distributed optical fiber sensing technology has also been used in engineering monitoring because of its flameproof, explosion-proof, corrosion resistance, electromagnetic interference resistance, high-pressure resistance, remote measurement, real-time measurement, and positioning characteristics [18,19,20]. The distributed optical fiber temperature measurement technology has been widely used in leakage detection of geotechnical structures in recent years and has new applications in monitoring oil well temperature [21].

The scattering intensity of the optical signal transmitted by the optical fiber is abnormal with the change in the ambient temperature. The abnormal temperature area can be calculated by optical time domain reflection. The distributed optical fiber temperature sensing system (DTS) was used for demodulating the feedback optical signal to obtain the real-time temperature of the abnormal position, which is equivalent to a linear sensor [22]. DTS was developed by New York Sensors Limited, a British company. German GTC Company used this equipment to conduct a series of studies on the application of dam bodies [23,24]. Bekele et al. [25] proposed a DTS technique based on Brillouin scattered light to monitor dam seepage. In the Three Gorges Project of China, an optical fiber temperature measurement technology was applied to the No. 14 dam section of the Left Plan. The buried optical cable was an armored temperature-measuring optical cable, which was not easily damaged. The test results showed that the dam section is relatively healthy, no leakage area has been found, and it can operate normally and run stably [26]. Nearly 6 km of temperature-measuring optical cables are buried in the concrete of the dam body of the Lancang River Jinghong Power Station of China, which was the largest scale of optical cable deployment at that time [27]. Wu [28] conducted an experiment on the relationship between temperature difference and seepage velocity of optical fiber. They concluded that a quadratic relationship existed between them. Zeng et al. [29] heated the distributed optical fiber arranged in the tunnel with a strong current, making its average temperature rise about 4.5 °C, the leakage location of the tunnel is located according to the phenomenon that the temperature rise of some parts along the optical fiber is obviously lower than the average. Yaser [30] achieved laboratory-scale continuous monitoring of seepage through a passive optical fiber distributed temperature sensing (DTS) system, and the seepage process and heat transfer process of the sand body model are analyzed numerically. Wang et al. directly measured the temperature field distribution inside the dam with the temperature gradient method, the closer the distance to the leakage channel, the more obvious the temperature difference, the leakage site was located by analyzing the temperature field measured by optical fiber. The above research shows that the distributed optical fiber temperature measurement technology is very mature. In recent years, many scholars have used this technology as a theoretical basis for applying optical fiber as a linear sensor to engineering leakage detection, and they have achieved many research results [31]. However, the application of optical fiber temperature measurement technology to the curtain leakage detection of foundation pits has not too many precedents, particularly for ultradeep foundation pits.

Leakage of the underground engineering was an important safety problem, which may cause internal erosion of the structure. At present, the traditional detection method was still the most commonly used detection method in underground engineering. The traditional detection method of water leakage in underground engineering referred to the qualitative judgment of water leakage by means of manual inspection, visual inspection, percussion, photography, drilling, and data searching. However, traditional detection methods have the following disadvantages.
(1)The detection efficiency was low, the workload was large, and the information feedback cycle was long.(2)The accuracy of the test results was poor and the subjectivity was strong, which required the theoretical background and experience of the test personnel.(3)The detection results were limited to the surface of the structure and could not reflect the water leakage inside the structure.(4)The detection time was narrow, so the detection time should be staggered with operation hours.

Therefore, the traditional seepage detection method of underground engineering cannot meet the needs of engineering. In order to overcome the shortcomings of traditional detection methods of seepage in underground engineering, researchers have developed a variety of rapid nondestructive detection technologies of seepage in underground engineering based on different theoretical principles. Among them, distributed optical fiber sensing technology based on the temperature tracer method has attracted more and more attention due to its unique advantages of high sensitivity, good stability, and high resolution.

Based on the No. 4 foundation pit of the Guangyuan project, the characteristics of the temperature measurement data of the temperature-measuring optical cable were analyzed through simulation tests. Moreover, the corresponding relationship between the abnormal temperature measurement of the optical cable and the leakage area was studied. A set of implementation schemes for pit leakage detection was designed, and the optical cable was laid according to the scheme on the construction site to test the feasibility of the scheme (pipeline survival rate and construction difficulty). The superposition of the temperature field and seepage field during curtain leakage was simulated by a numerical method. Moreover, the variation in the temperature field distribution with time was analyzed. The active temperature field leakage detection could detect the heating time required when the temperature was abnormal.

## 2. Conceptual Model

Active temperature field leakage detection was the method of heating the layers and groundwater around the joint of the earth-facing surface of diaphragm walls through artificial heating. During the process, the water temperature around the joint outside the wall was high. Before the excavation of the foundation pit, the soil outside the wall was quickly heated when the foundation pit was not dewatered. The inside of the wall would not be substantially heated because of the poor thermal conductivity of the concrete wall of the reinforced concrete. The inside of the wall slowly heated after a period of heating. If the wall joint had defects, the water inside would be connected with the water outside the wall at the joint. At this time, the heated groundwater flowed into the wall joint, and the abnormal temperature rise could be detected in the wall. During the foundation pit excavation, the groundwater level in the foundation pit needed to be lowered. If the seam had a defect, the groundwater outside the wall would flow into the wall. At this time, the temperature-measured optical cable in the wall could detect that the leakage area was slow. The water outside the wall kept flowing into the pit, and the heating efficiency of the heating system outside the wall was limited. Thus, the effect of active temperature field leakage measurement was remarkable in the dewatering process of the foundation pit after the diaphragm wall construction. Manual heating uses the method of circulating hot water on the heating pipeline. The steel pipe was fixed on the steel bar near the earth-facing surface of the steel cage on both sides of the underground diaphragm wall joint. The steel pipe was fixed on the reinforced noodles on both sides near the underground diaphragm wall seams, as shown in Figure 1a.

The passive temperature field leakage did not change the temperature of the surrounding environment of the diaphragm wall seams. Instead, the optical fiber itself was heated using a self-heating armor temperature cable. During the excavation of a foundation pit, the water outside the wall flowed into the pit from the leakage position. The heating optical cable set near the leakage location of the seams caused low temperatures and abnormalities because of the heat from the water, as shown in Figure 1b. The effect of the passive temperature field leakage test during precipitation in the foundation pit excavation was remarkable.

## 3. Distributed Optical Fiber Leakage Simulation Test

### 3.1. Test Instruments

#### 3.1.1. Temperature Measuring Instrument

The FlukeTiR110 infrared thermal imaging instrument is used to measure and collect the temperature data field of the heating pipeline, as shown in Figure 2. FlukeTiR110 infrared thermal imager is a handheld, infrared thermal imaging camera. The temperature range of measurement is −20~+150°, the accuracy is ±2°, and the minimum focusing distance is 15 cm. Its imager displays thermal images on a high-definition LCD screen, and Images can be saved to an SD memory card.

#### 3.1.2. Optical Fiber Temperature Measurement System

The Sentinel-DTS distributed fiber temperature monitoring system (as shown in Figure 3a) used in this article is produced by Shanghai Bohui Communication Technology Co., Ltd. The system utilizes an optical fiber sensing signal and transmission signal simultaneously. This system adopts the characteristics of advanced OTDR technology and Raman scattered light to temperature sensitivity, which can detect changes in temperature along different positions of optical fibers and achieve truly distributed measurement. It is very suitable for a variety of long-distance temperature measurement. The temperature resolution of the DTS temperature measurement system can reach 0.01 °C, and any small temperature change can be detected. The maximum test distance is 30 km, and the minimum spatial resolution is 0.1 m.

The DTS system uses a special temperature-sensitive optical cable as a detector without charge, with the essence of explosion-proof, lightning protection, corrosion resistance, electromagnetic interference, and other advantages. The DTS temperature measurement system has been widely used in oil, pipeline leakage, tunnel fire detection, dam safety monitoring, power cable monitoring, and so on.

Armored temperature-measuring optical cable: The model of armored temperature measurement optical cable is SCJKBH-1A1b-30-R, and the main component of optical fiber is quartz. This optical cable was wrapped in stainless steel sheets around the quartz fiber and tightly wrapped in Kaifra. This approach ensured that the optical cable was not easily broken or that 90° bending could not easily cause optical fiber damage. The flame retardant protection cover in the outer layer is shown in Figure 3b.

Heating-type armored temperature-measuring optical cable: The type of heating armored temperature measuring cable is SCTGTCY-2A1b + 2B1 + 22C − 5.0 * 8.5-BL, and the main component of optical fiber is quartz. This optical cable is a sensing optical cable designed for the liquid leakage monitoring application of large buildings, such as dams and pipelines. The optical cable could be buried directly during the pouring of concrete. As shown in Figure 3c, the optical cable comprised a PE outer cover with a high-intensity center beam tube, built-in sensing fiber, water-resistant filling, insulating layer, and parts enhanced by the embedded heating copper wire. The optical cable exhibited good bend-resisting scraping and signal transmission performance, which can be widely used in engineering testing.

### 3.2. Direct Device Ground Connection Test

#### 3.2.1. Ground Heating System Connection

Ground heating tests were performed to test the applicability of the heating pipeline and the stability of the heating system and check whether the heating system could run stably. The time required for heating to the constant temperature, pipe sealing performance, the sensitivity of DTS instruments, and the stability of the fiber temperature measurement system were determined. In this study, a loop was formed on the ground. The total length of the pipeline was 120 m. The cable extended 40 m from the outlet of the electromagnetic heating furnace. The installation of on-site temperature-measuring optical cables and data collection instruments is shown in Figure 4. Temperature-measuring cables could be laid around the heating pipe to detect the abnormal temperature in the extension direction of the heating pipe after the ground heating system was laid and to test the detection effect of the temperature-measuring cable and DTS system.

The heating system was heated by the frequency conversion electromagnetic heating furnace (Figure 4a), which was connected to the 1.2-inch galvanized steel pipe for circulating hot water to heat the surrounding environment. The steel pipe used in this article is a 1.2-inch hot-dip galvanized steel pipe with an outer diameter of 41 mm, which is produced by Zhejiang Jinzhou Pipeline Technology Co., Ltd., Huzhou, China, The heating system included insulation water tanks (Figure 4b), 320 W supercharged water pump (Figure 4c), 1.2-inch heat-galvanized steel pipes, 1.2-inch stainless steel ripples (steel pipe connections at the turning point), connecting parts and. microcomputer variable frequency electromagnetic heating water stove.

The test site was equipped with a secondary electrical box and a third-level electric box to meet the power supply demand of an electromagnetic heating furnace, supercharged water pumps, DTS, and other electrical appliances. All the connection parts were connected with a thread. Two galvanized steel pipes were equipped with a live connection, and the pads were placed in the middle. After the threaded wire was wrapped in the raw material, it was tightened to prevent the pipeline from leaking.

The heating system could work normally after debugging and improvement. The maximum water temperature in the heating furnace could reach 75 °C to 80 °C, and the circulating water temperature in the measured pipe could stabilize between 65 and 70 °C. During the heating system and constant temperature operation, the surface temperature around the heating pipe and electromagnetic heating furnace was scanned by the infrared thermal imager, as shown in Figure 5 (the ambient temperature during the test period was 22.4 °C). Figure 6 shows that the temperature near the electromagnetic heating furnace was higher than that of the surrounding atmosphere. The temperature of the heating system could reach 55 °C. The maximum temperature reached 100 °C because of the water vapor near the exhaust valve.

#### 3.2.2. Simulated Leakage Condition

This study simulated the abnormal waveform data of DTS under single-section insulation, multi-section insulation, single-section leakage, and multi-section leakage conditions. The shape of the heat source corresponding to abnormal data was summarized. The multistage alarm was designed in DTS to test the sensitivity of the temperature alarm.

(1)Abnormal detection of multistage heating and cooling

The heating pipeline and optical cable were wrapped in a specific area to reduce the heat loss in this section and detect the abnormal temperature increase in the optical fiber during the heating process and the cooling law after the heating was stopped. Before the start of heating, the high-temperature alarm value was set to 40 °C. The insulation foam was wrapped 9 m to 10 m away from the DTS interface to simulate single-stage insulation as shown in Figure 6. The self-adhesive insulation cotton was wrapped 11–12 m, 28.5–29.5 m, and 36.5–37.5 m to simulate multistage insulation, as shown in Figure 7. The temperature variation law of the optical cable during the heating and cooling processes of the pipeline was detected. The test conditions are shown in Table 1.

(2)Multi-section abnormal temperature detection

After the whole pipeline was kept at a relatively high temperature, the electromagnetic heating furnace was in heat preservation mode. Cold water was poured into the heating pipe in a specific area, and the water temperature was 16 °C. Artificial watering was used to simulate groundwater leakage, and the abnormal law of temperature drop during the constant temperature operation of the optical cable was detected. The maximum heating temperature of the electromagnetic heating furnace was set to 75 °C during the heating process, and the DTS high-temperature alarm value was set to 70 °C. First, the constant temperature distribution that the optical fiber could achieve was detected before the constant temperature anomaly detection test (watering at a specific location). If the temperature distribution was uneven, constant temperature anomalies would appear in local areas, indicating that the cable at this location was not firmly fixed with the heating pipeline or the heating pipeline here leaked. The test could only begin after investigation and repair according to the actual situation.

In the case of a single-stage seepage, continuous water at positions 14–15 m and 20–21 m away from the optical cable extension of the DTS interface was first simulated. The temperature distribution of the optical cable was observed. Then, the multistage leakage was simulated simultaneously, with water at 17–18 m and 20–21 m and 17–18 m and 25–26 m. The temperature distribution of the optical cable was observed. The test conditions are shown in Table 2.

### 3.3. Analysis of Test Results

#### 3.3.1. Analysis of DTS Temperature Alarm Sensitivity and Temperature Measurement Characteristics

During the pipeline heating process, the alarm value was first reached at the position 9.5 m away from the DTS interface after approximately 5 min of heating when the high-temperature alarm was set at 40 °C. The current temperature reached 40.2 °C. The reason for the fast temperature increased at this position was that the heat-insulating foam was wrapped outside the pipe and optical cable. Moreover, the heat dissipation was reduced, as shown in Figure 8.

The high-temperature alarm in this section was canceled, the insulation foam and continuous heating were removed, the heating temperature was adjusted to 75 °C, and the high-temperature alarm temperature was set to 70 °C. As shown in Figure 9, the alarm value was first reached at a place 19 m away from the DTS interface after approximately 30 min of heating. The temperature at this point was 70.2 °C.

The 70 °C alarm value was canceled, and the heating furnace entered the 75 °C heat preservation mode. At this time, the temperature of the pipeline was stable at approximately 65 °C, as detected by the optical cable. Moreover, the maximum temperature difference between different detection points was 6 °C. These findings indicated that the optical cable could obviously detect the abnormal leakage and that the temperature difference between the leakage area and the environment needed to be greater than 6 °C. The measured temperature image is shown in Figure 10.

#### 3.3.2. Analysis of Abnormal Temperature Rise and Fall

The temperature rise rate of the wrapped insulation cotton was obviously fast shortly after the heating started. The pipeline temperature tended to stabilize after heating for 40 min. The measured temperature was recorded every 5 min during the heating process, and the temperature change curve was drawn, as shown in Figure 11.

As shown in Figure 11, the temperature monitored by the optical fiber increased with the heating time. The longer the heating time was, the slower the growth trend of the pipeline temperature was. The average temperature of the pipe was 45.9 °C when heated for 10 min, and the temperature in the insulation-cotton-wrapped area was 49.3 °C. When heated for 20 min, the pipeline had an average temperature of 50.2 °C, with the highest temperature of 60.7 °C. When heated for 30 min, the pipeline had an average temperature of 60.3 °C, with the highest temperature of 66.6 °C. The maximum temperature difference across the pipeline was only 2.5 °C when heated for 35 and 40 min. Figure 11 shows that the temperature in the area wrapped in thermal insulation cotton could reach a maximum of 74.6 °C when the overall temperature of the pipeline was heated to 67 °C. Moreover, the abnormal temperature could be clearly identified. The power supply was cut off after the temperature stabilized. The abnormal temperature variations were monitored in the natural cooling process. The measured temperature was recorded every 5 min. The temperature change curve is shown in Figure 12.

Figure 12 shows that the temperature detected by the optical fiber decreased with the increases in cooling time. The temperature variation of the pipeline was remarkable when the temperature dropped for 5 min. When the temperature was cooled for 10 min, the pipe temperature decreased by 1.1 °C on average, whereas the temperature in the insulation-cotton-wrapped area only decreased by 1.2 °C. However, the temperature variation range of the pipeline was small when the cooling time was within 20–25 min. Moreover, the average temperature of the pipeline was 1.9 °C. The temperature of the pipeline was constant when the cooling time exceeded 40 min. At this time, the average temperature of the pipeline was 22.5 °C, and the temperature of the insulation-cotton-wrapped part was 16.3 °C. Figure 12 shows that the temperature at the position wrapped by thermal insulation foam decreased and lagged during the cooling process. The detected abnormal temperature points had a good correspondence with the actual thermal insulation area.

#### 3.3.3. Analysis of Abnormal Constant Temperature

The water temperature in the pipeline was stable between 64 and 69 °C after the electromagnetic heating furnace was heated to 75 °C and maintained at a constant temperature operation. At this time, the leakage simulation test was performed. Single-section watering was performed for different parts of the pipeline. The temperature curves of the pipeline were obtained, as shown in Figure 13 and Figure 14.

Figure 13 shows that after continuous watering at a distance of 22–23 m from the DTS joint (the first 8 m of the optical cable was not laid on the heating pipe), the maximum temperature dropped approximately 11.6 °C within the range of 22–24 m. Moreover, the minimum temperature was 55.6 °C, which occurred at 23 m. Figure 14 shows that after stopping the watering and waiting for the temperature to stabilize, the maximum temperature dropped by approximately 18 °C within 28–30 m after continuous watering at a distance of 28–29 m from the DTS joint. The minimum temperature was 48.7 °C, appearing at the position of 29 m.

For different parts of the pipe in multiple sections, the temperature curve of the pipe is shown in Figure 15 and Figure 16.

The temperature dropped after watering at 25–26 m and 28–29 m away from the DTS joint are shown in Figure 15. In the range of 25–27 m, the maximum temperature dropped to 18.2 °C, and the minimum temperature was 47.8 °C, which appeared at the position of 26 m. In the range of 28–30 m, the maximum temperature dropped to 18.8 °C, and the minimum temperature was 47.9 °C, which occurred at the position of 29 m. Watering was stopped, and the temperature on the optical cable line returned to stability. Figure 16 shows that in the range of 25–27 m, the maximum temperature dropped to 20.6 °C, and the minimum temperature was 45.4 °C after watering at 25–26 m and 33–34 m away from the DTS joint, which appeared at the position of 26 m. In the range of 33–35 m, the maximum temperature dropped to 19.1 °C, and the minimum temperature was 45.7 °C, which occurred at the position of 34 m. In general, the artificial cooling area and the abnormal temperature area measured by the optical cable had good correspondence. The error was within 1 m, and the abnormal temperature difference anomaly ΔT was approximately 18 °C, which was obviously greater than the maximum temperature difference of the line itself ΔT. The detection effect was noticeable.

## 4. Numerical Experimental Analysis of the Active Temperature Field Leak Detection Method

The active temperature field leakage detection arranged a heating system outside the wall to detect abnormal temperature conditions near the joints inside the wall. The heating system and the temperature-measuring optical cable were bound and fixed on the steel cage surface. They were wrapped in the concrete protective layer after the concrete was poured because of the limitations of the construction technology. If the joints of the diaphragm wall had cracks, the underground confined water would flow from the outside of the wall to the inside of the wall. In this study, COMSOL Multiphysics software was used to establish the temperature field and seepage field for superposition simulation and study the heating effect of the heating system, that is, the influence of the heating system outside the wall on the temperature field inside the wall.

### 4.1. Principles of Numerical Modeling

#### 4.1.1. Introduction to COMSOL Software and Conjugate Heat Transfer Module

The Comsol Multiphysics software was used for simulation. Comsol involves a large number of computing modules and can carry out coupling analysis of any field, which has an extremely wide range of applications. It can conveniently imitate the physical process in practical engineering, and has been applied to the existing relevant acoustics, electromagnetics, porous medium, fluid, and other physical simulation and coupling analysis. The software is based on the finite element method, combined with the function definition relationship of actual physical parameters, and the parameters are expressed and defined by the differential equation, so as to carry out the numerical simulation of physical phenomena. For the coupling analysis of multiple physical fields, the core equation of Comsol software is a partial differential equation, through which the coupling analysis of different physical fields can be carried out. Therefore, the physical parameters and phenomena that can be expressed by partial differential equations can be analyzed by Comsol software.

At the heat transfer interface, calculations are made based on the law of conservation of energy, the laminar flow interface is calculated based on the two-dimensional seepage differential equation, and the Kays–Crawford model is used in non-temperature flow.

Conjugate heat transfer describes the heat transfer in solids and liquids and nonisothermal flow in liquids. The main heat transfer mechanism in solids is heat conduction, whereas that in fluids mainly includes heat conduction and convection. Based on the energy conservation law, this module considers energy conservation in the process of conduction, convection, and radiation to calculate the temperature field change. It can combine the heat transfer process with the fluid flow through multiphysics field coupling.

In the Conjugate heat transfer model, solid heat transfer conforms to the law of conservation of energy. In this paper, the wall heat transfer equations were
(1)$$d_{Z}\rho C_{p}\frac{\partial T_{2}}{\partial t}+d_{Z}\rho C_{p}u\cdot \nabla T_{2}+\nabla \cdot q=d_{Z}Q+q_{0}+d_{Z}Q_{ted}$$
(2)$$q=-d_{Z}K\nabla T_{2}$$
where *d_z_* is the thickness, *ρ* is the density of permeable water, *C_p_* is the specific heat capacity of the liquid, *T* is the temperature of the solid, *t* is time, *u* is the flow velocity vector, *q* and *Q* are both heat sources, ∇ is Hamiltonian operator, *K* is permeability coefficient.

The soil heat transfer equation was
(3)$$d_{Z}\rho C_{p}\frac{\partial T_{2}}{\partial t}+d_{Z}\rho C_{p}u\cdot \nabla T_{2}+\nabla \cdot q=d_{Z}Q+q_{0}+d_{Z}Q_{p}+d_{Z}Q_{ted}$$
(4)$$q=-d_{Z}K\nabla T_{2}$$

The hypothesis equation of laminar flow was
(5)$$\rho \frac{\partial u}{\partial t}+\rho \left(u\cdot \nabla \right)u=\nabla \left[-pl+\mu \left(\nabla \mu +\left(\nabla \mu \right)T\right)-\frac{2}{3}\mu \left(\nabla u\right)l\right]+F$$
(6)$$\frac{\partial \rho }{\partial t}+\nabla \cdot \left(\rho u\right)=0$$
where *p* is the pressure, *l* is the length of the underground diaphragm wall joint.

#### 4.1.2. Numerical Simulation in Actual Working Conditions

The numerical simulation mainly analyzed the variation of the temperature field around the wall with time under constant temperature heating of the heating system. The heating pipe was arranged in the concrete protective layer of the outer wall. The temperature-measuring optical cable was arranged in the concrete protective layer of the inner wall. The heat transfer in the concrete wall was solid, and heat conduction was mainly considered. If the wall had cracks, the heat would be conducted through the water flow in the soil, including heat conduction and heat convection. The fracture and the surrounding water body were in a free-flowing state before the foundation pit was dewatered. Thus, the laminar flow interface was used for calculation in the conjugate heat transfer module. The conjugate heat transfer module consists of three interfaces: heat transfer interface, laminar flow interface, and nonisothermal flow interface.

#### 4.1.3. Establishment of the Model

The cage reinforcement included longitudinal main reinforcement and lateral tie bars. Two sections with tie bars and without tie bars should be selected for research when selecting horizontal sections because of the good thermal conductivity of steel bars. In the underground diaphragm wall model, the thickness of the underground diaphragm wall and the spacing, diameter, and quantity of the steel bars were consistent with the engineering design scheme. Two kinds of two-dimensional profile models were established. The connection position of the underground diaphragm wall was plain concrete with a thickness of 30 cm. Moreover, a risk of leakage existed. The crack width was artificially set in the plain concrete to simulate the actual curtain leakage. The fracture width was assumed to be 10 cm. The constructed two-dimensional section model is shown in Figure 17a,b.

Subsequently, assign material attributes to geometric models in the material library. There is no physical parameter of soil around the foundation pit in the material library, so it is necessary to create new materials and input physical properties such as thermal conductivity, density, constant pressure heat capacity, dynamic viscosity, specific heat rate, etc. A conjugate heat transfer module was added after parameter assignment, heat transfer module, laminar flow module, and non-isothermal flow module were defined respectively, and set the boundary conditions and initial values. Take the model with tension reinforcement as an example.

(1)Heat transfer interface

The heat transfer equation conformed to the conservation of energy, but the mathematical equation of heat transfer in the wall and soil was different. The upper and lower boundary of the wall was an insulating thermal field, while the surrounding soil was an open thermal field. Throughout the system, the initial temperature field was the same, and the heat source was set in the installation area of the heat-conducting steel pipe. See Figure 18.

(2)Laminar flow interface

In the laminar flow interface, as shown in Figure 19a, only the soil mass and the joint of the diaphragm wall had the property of the fluid. As shown in Figure 19b, before calculation, the hydraulic gradient of the whole system was the same. As shown in Figure 19c, open boundary conditions existed at the soil boundary, the contact surface between wall and soil, and the joint of the diaphragm wall.

(3)Non-isothermal flow interface

At the non-isothermal flow interface, no boundary conditions and initial values were adopted. As shown in Figure 20, non-isothermal flow can occur at the joint between the soil mass and the underground diaphragm wall.

Then, the grid was divided, as shown in Figure 21, there were 236 vertices, 4395 boundary elements, and 57,917 blocks. After the meshing was completed, the transient solver was selected for calculation. The transient study could realize the calculation of temperature field changes with time. Finally, the visual calculation results are output.

### 4.2. Numerical Simulation Results

The conjugate heat transfer interface was used to calculate the temperature field distribution of the above two models. The temperature inside the wall increased accordingly after the heating system outside the wall continued heating for a time. When the temperature difference between the location where the optical cable was laid in the wall and the ambient temperature was large (>ΔT = 6 °C), the confined water outside the wall would continue to flow into the pit if purdah leakage occurred during dewatering. Apparent low-temperature anomalies at the leakage location could be observed. Therefore, the boundary of the soil in the laminar flow interface was set as an open boundary; that is, the water in the soil was in a free-flowing state, the ambient temperature was set to 20 °C, the temperature of the heating pipe was set to 80 °C, and the heat transfer temperature for 0–50 days was studied. The temperature field variations with time are shown in Figure 22 and Figure 23.

The distribution of the temperature field for 1, 5, 10, 20, 35, and 50 days of heating above showed that the temperature of the underground diaphragm wall and the surrounding soil increased continuously with the increase in heating time. Moreover, the heat propagated faster in the soil outside the underground diaphragm wall than in the underground diaphragm wall. The temperature propagated quickly at the section position with tension reinforcement because of the good thermal conductivity of the steel bar. The position with tension reinforcement in the wall heated at the same position at the same time had a high temperature.

The minimum temperature of the measuring point inside the wall was required to be greater than 6 °C above the ambient temperature. Therefore, in this study, the longitudinal section line at the position of abscissa x = 1.13 m (inside the concrete protective layer inside the wall) was selected, as shown in Figure 24. The temperature curves (heating for 0, 5, 10, 15, and 20 days) are shown in Figure 25. Figure 26 shows the temperature variation curve of the cable placement position (1.13, 0.19) within 25 days of heating.

Figure 24 shows that the temperature of the surrounding environment before the detection was 20 °C. As the buried depth increased, the temperature curve varied in a wave shape. The detected temperature increased with the increasing heating time. However, the increase rate decreased gradually. Figure 25 and Figure 26 show that if the heating was continued for approximately 10 days, the temperature of the measuring point inside the wall could increase by 6 °C. At this time, abnormal leakage of the curtain could be detected. The temperature of the measuring point reached approximately 30 °C after heating for 20 days. At this time, the lowest temperature in the concrete protective layer was nearly 27 °C, and abnormal temperatures could be measured at all positions.

## 5. Field Operation of Distributed Optical Fiber Leak Detection in Foundation Pit Engineering

### 5.1. Site Construction Overview and Engineering Geological Conditions

This test relied on the No. 4 working well of the Guangyuan Project. The geographical location of this project. The excavation depth of the foundation pit was approximately 42 m, and the column piles and uplift piles were set under the base. The length of the piles was approximately 40–50 m, the depth of the underground diaphragm wall was 86 m, and the length of the steel-reinforced cage was 85.45 m. The underground diaphragm wall was not allowed to have any leakage. The plane size of the No. 4 working well was 56 m × 56 m, the diaphragm wall thickness was 1.2 m, and the depth of buried ground was approximately 86 m. The diaphragm wall was divided into a first-stage trough and a second-stage trough. The two troughs were connected by a milling joint. A total of 52 diaphragm walls existed.

The ground of the project site had a slight undulation. The overall elevation was between 3.33 and 7.88 m, and the height difference was 4.55 m. The project crossed the Chuanyang River, a tributary of the Huangpu River. The surface water system was developed. The average buried depth of the water level was 1.94 m, and the water level depth was often affected by atmospheric precipitation and fluctuation. The confined water layer near the project site was mainly composed of gray clayey silt mixed with silty clay and gray sandy silt mixed with silty clay. Most of the area of the No. 4 working well was directly connected with the first confined aquifer below.

### 5.2. Design of Leakage Detection Scheme for Diaphragm Wall

This test was constructed on the second-stage grooved reinforcement cage. The total length of the reinforcement cage was 85.5 m, and the construction was divided into three sections. The length from bottom to top was 37.5 m + 38 m + 10 m, respectively. The reinforcement cage was first welded in three sections on the template and then hoisted into the second-stage trough in sections. The heating pipe should also be fixed on the steel cage made on the template to ensure that the steel cage would not fall off during the hoisting process. The two heating pipes were connected during the butt joint process of the steel cage. After the optical cable was fixed at the bottom section of the steel-reinforced cage, the other two sections of steel-reinforced cages were tied up while being lowered after the section of the cage was lowered.

The heating pipes were 1.2-inch galvanized steel pipes and six-point PE-RT and PPR plastic pipes to determine the pipeline layout method and the heating pipeline material suitable for the diaphragm wall construction process and ensure that the pipeline could run stably after the diaphragm wall was formed. The steel pipes were connected by thread union and hot welding of plastic pipes. The optical cables were armored temperature-measuring optical cables and heating-type armored temperature-measuring optical cables. The pipeline layout scheme in the three reinforcement cages is shown in Table 3.

The layout procedure of the heating pipe and temperature measuring optical fiber is shown in Figure 27. The heating pipe is arranged on the earth-facing surface. Since the heating pipe needs to form a loop, it is also necessary to connect the loop at the bottom of the first steel cage. The two steel pipes were fixed with a live joint, and the U-shaped steel bar was used to clamp the movable joint and weld it on the main reinforcement of the steel cage to fix the heating pipe. Temperature-measuring optical cables were arranged on the earth-facing surface and pit-facing surface. The cable is bound to the main bar of the steel cage with a cable tie. A sufficient length of optical cable was reserved on the top of the steel cage to ensure that it can be exposed to the ground, and the excess optical cable was coiled on the top of the steel cage.

The first section of the steel cage was hoisted into the trough, and the cable on the remaining two sections of the steel cage was tied and fixed during the process of placing the steel cage. After the steel cage was lowered fully, the heating pipe interface shall be shielded to prevent the concrete from blocking the pipe mouth during the concrete pouring. After all the reinforcement cages are lowered, cover the position of the heating pipe joint to prevent the concrete from blocking the pipe mouth during the pouring of concrete. At the same time, the reserved length of the cable tray was placed near the guide groove for connecting detection instruments.

### 5.3. Engineering Application Test Results

(1)Survival condition of heating pipe: A galvanized pipe was used in Test 1, and the sectioned point were connected by stainless steel corrugated pipes. The sectioned point had water leakage. The steel pipe and the reinforcement cage were welded and fixed by U-shaped steel bars. Some parts of the steel pipe were welded to cause water leakage. In the process of lifting the steel cage, the mud entered the pipeline through the leakage position because the groove section was filled with mud wall protection. Moreover, the mud solidified after the concrete was poured, and the pipeline was blocked. In Test 2, the PPR and PR-RT plastic pipes had good overall sealing. The joints were fused to facilitate construction and prevent leaks.(2)Heating device improvement: PE-RT and PPR plastic pipes were used as heating pipes. The joints were fused to achieve a good sealing effect. A galvanized steel pipe was used. All pipes were connected by steel pipe wire, and the joints were tightly wrapped with waterproof tape to prevent water leakage. The steel pipe was bound and fixed with iron wire to avoid welding in the whole process (Test 3).(3)Survival of optical cable (Test 1): The optical cable was connected to form a loop at the bottom of the steel cage. The armored temperature-measuring optical cable was laid outside the wall. The armored temperature-measuring optical cable and the heating temperature-measuring optical cable were laid inside the wall. The total length of the optical cable was approximately 220 m. Both ends of a single optical cable loop could be connected to DTS detection. The test results of optical cable connectivity are shown in Table 4.

The cable for temperature measurement outside the wall was broken in many places. The actual measurement depth was approximately 10 m. The temperature-measuring optical cable and the heating optical cable in the wall could be measured normally. Only one fiber break in the heating optical cable did not affect the temperature detection. The above phenomenon was mainly caused by the cable in the process of stretching or pouring concrete when the local impulse caused fiber breaking. Therefore, standardized operation during the laying process could prevent fiber breaking.

## 6. Conclusions

In order to overcome the shortage of traditional detection methods for the potential leakage in underground engineering. The conceptual model, experiment, and numerical simulation of diaphragm wall leakage detection using distributed optical fiber were performed. The following conclusions were reached:(1)The realization of distributed optical fiber temperature measurement required a temperature anomaly in the local position where the temperature-measuring optical cable passed. According to this characteristic, two detection schemes were designed: active temperature field leakage detection and passive temperature field leakage detection. The two methods were designed for foundation pit leakage detection.(2)The DTS temperature detection system can accurately detect abnormal temperature areas (leakage area and heat preservation area). The optical fiber temperature-measuring system could display the minimum temperature difference of 0.1 °C, the error between the measured leakage position and the simulated leakage position was less than 0.1 m. which met the requirements of leakage detection in engineering construction.(3)Unsteady conjugate heat transfer model is established. Through simulation, it is concluded that the heating system needed to be heated continuously for at least 10 days, and the temperature at the position where the temperature-measuring optical cable was laid in the wall increased by approximately 6 °C. At this time, the temperature abnormality at the leaking position of the curtain could be detected.(4)Compared with galvanized steel pipe, PPR and PR-RT plastic pipe has better-sealing properties and is not easy to leak. The breaking rate of armored fiber is low, but when the fiber outside the wall is broken, the temperature measuring cable and heating cable inside the wall can be measured normally.

## Figures and Tables

**Figure 1 materials-16-00561-f001:**
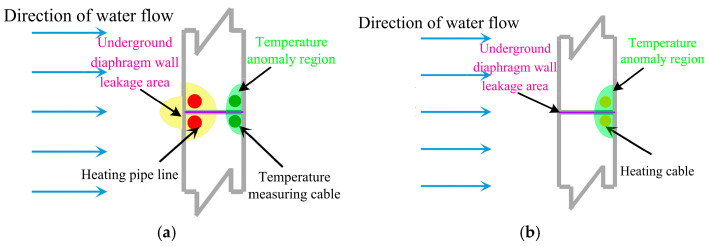
Active and passive temperature field leakage schematic diagram (**a**) Active temperature field leakage; (**b**) Passive temperature field leakage.

**Figure 2 materials-16-00561-f002:**
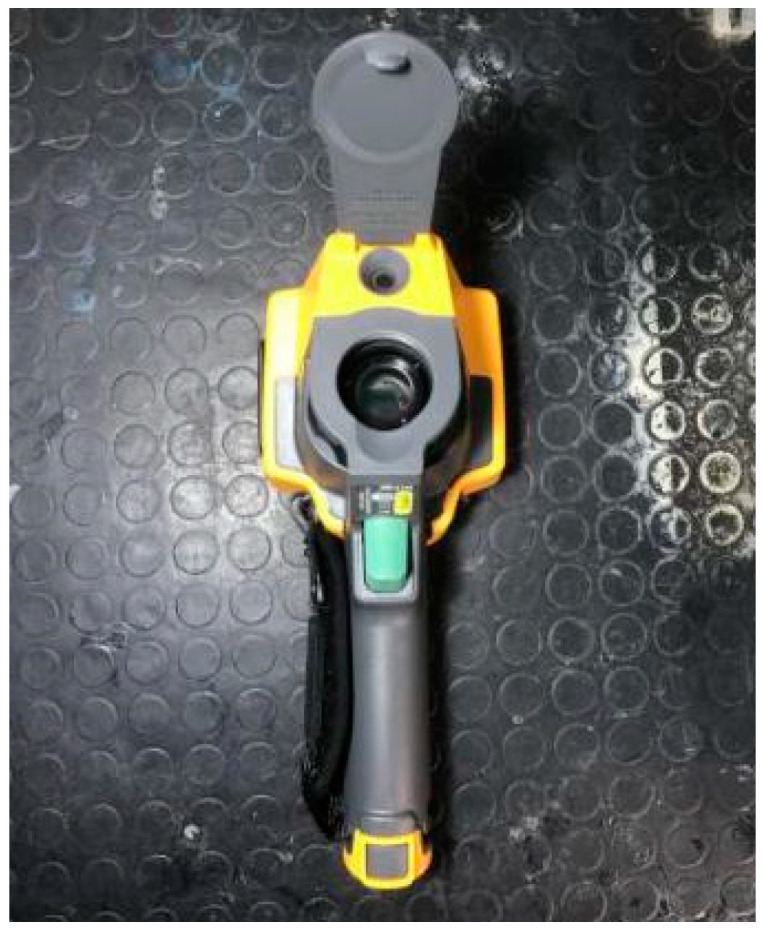
Infrared thermal imager.

**Figure 3 materials-16-00561-f003:**
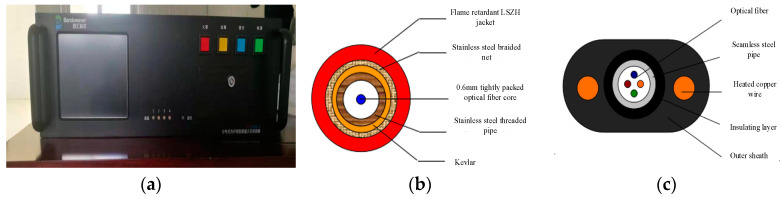
Optical fiber temperature measurement system; (**a**) DTS demolition instrument; (**b**) Armored temperature measuring optical cable; (**c**) Heating armored temperature measuring optical cable.

**Figure 4 materials-16-00561-f004:**
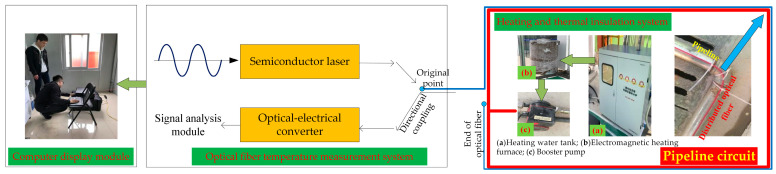
Connection of the ground heating system.

**Figure 5 materials-16-00561-f005:**
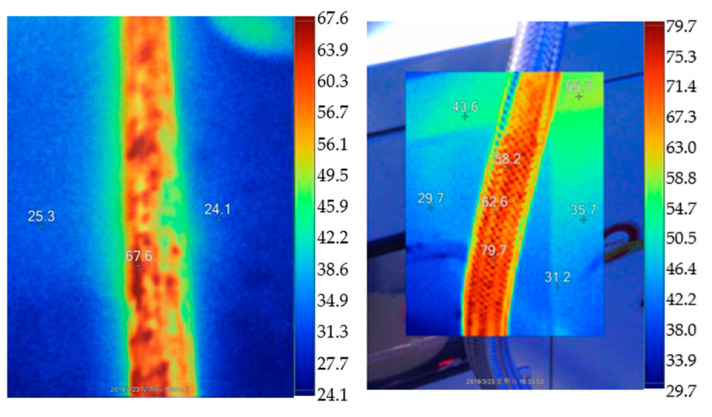
Temperature scanning of infrared thermal imager on the surface of the heating pipe and electromagnetic heating furnace.

**Figure 6 materials-16-00561-f006:**
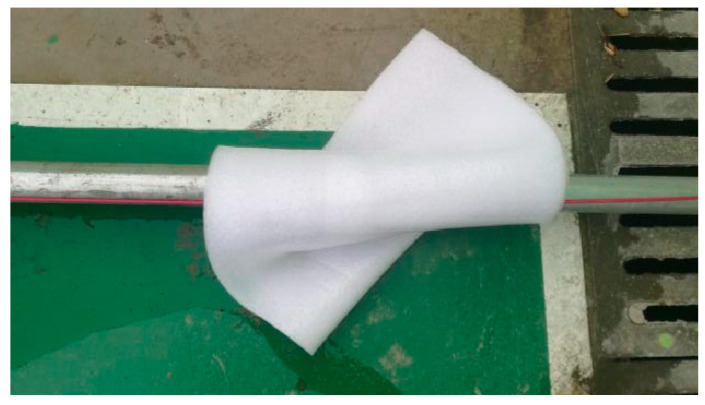
Steel pipe is covered with thermal insulation foam.

**Figure 7 materials-16-00561-f007:**
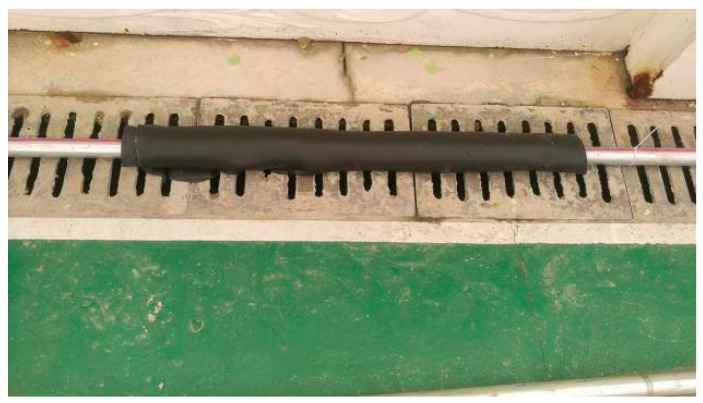
Steel pipe is covered with thermal insulation cotton.

**Figure 8 materials-16-00561-f008:**
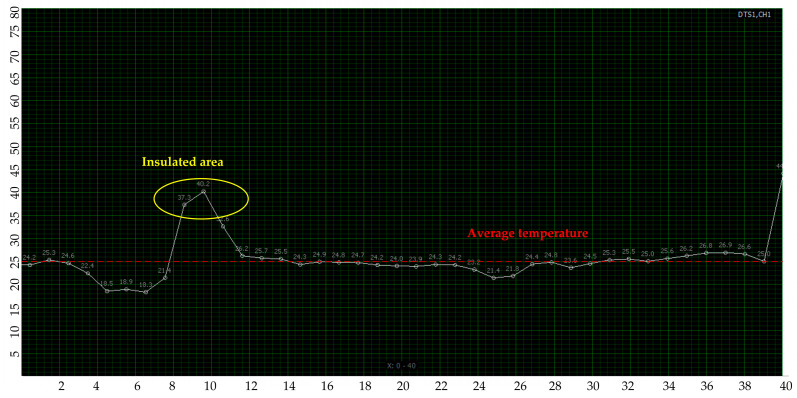
Temperature anomaly detection image.

**Figure 9 materials-16-00561-f009:**
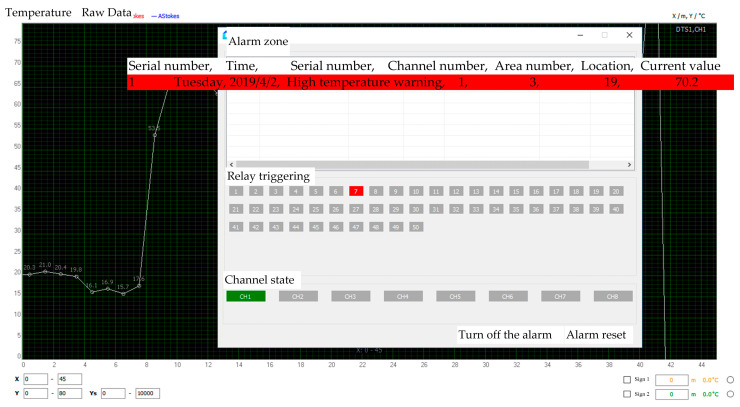
Interface image of DTS high-temperature alarm system.

**Figure 10 materials-16-00561-f010:**
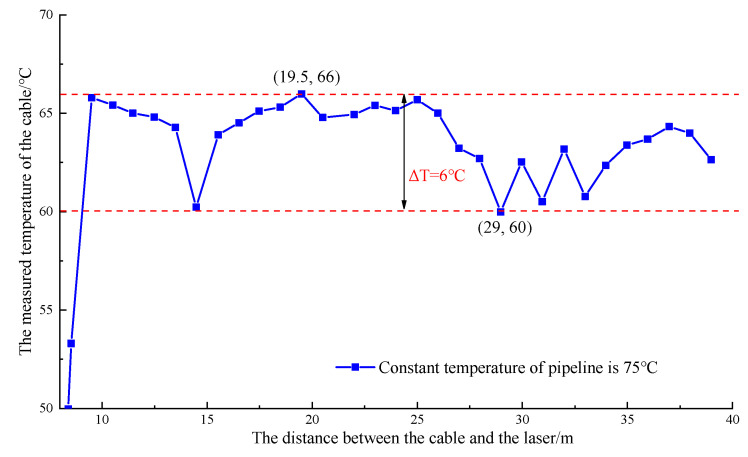
Image of pipeline constant temperature operation test.

**Figure 11 materials-16-00561-f011:**
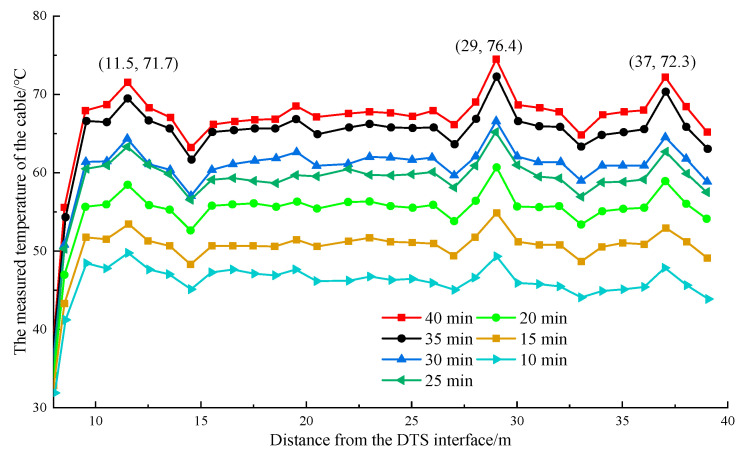
Temperature change curve of the pipeline.

**Figure 12 materials-16-00561-f012:**
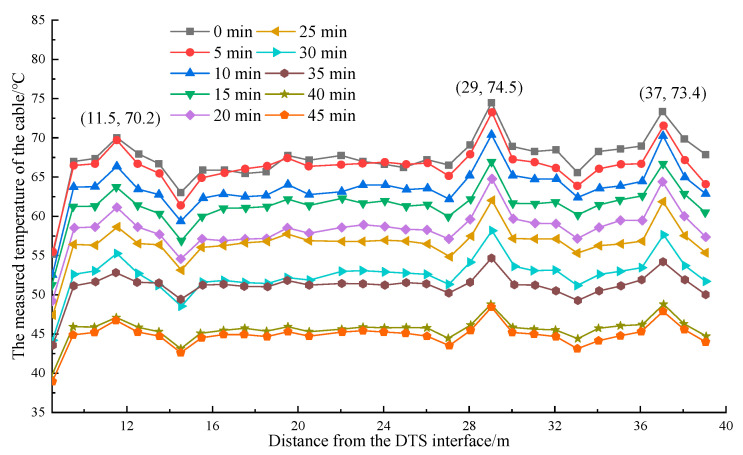
Variation curve pipe temperature.

**Figure 13 materials-16-00561-f013:**
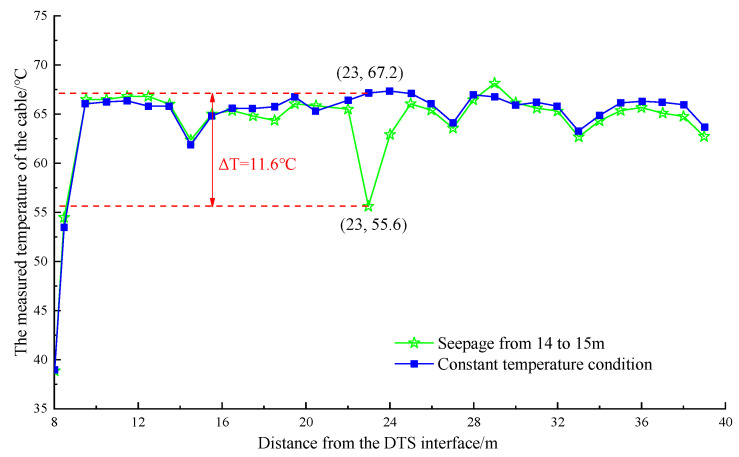
The temperature drop curve at the position of 14~15 m in the pipeline.

**Figure 14 materials-16-00561-f014:**
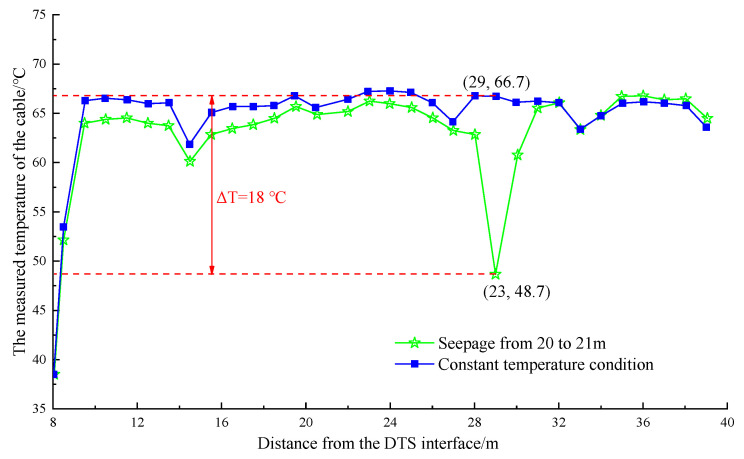
Temperature drop curve at the position of 20~21 m in the pipeline.

**Figure 15 materials-16-00561-f015:**
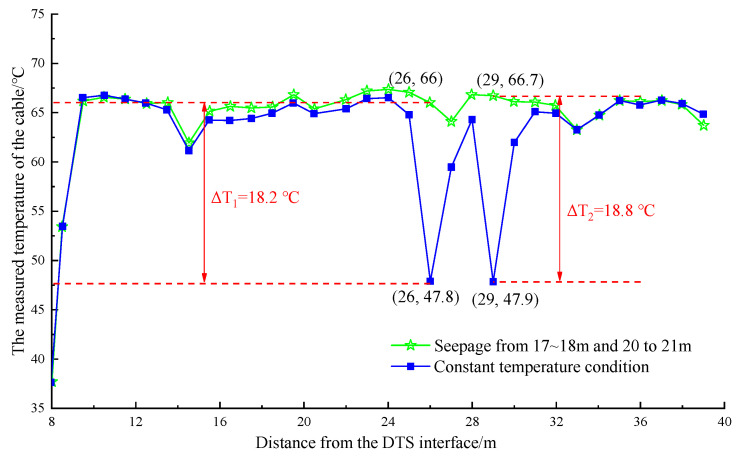
Temperature drop curve at the position of 17~18 m, 20~21 m in the pipeline.

**Figure 16 materials-16-00561-f016:**
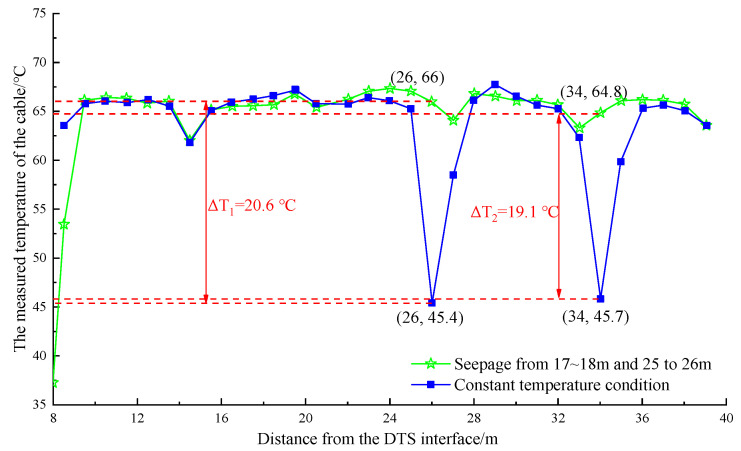
Temperature drop curve at the position of 17~18 m, 25~26 m in the pipeline.

**Figure 17 materials-16-00561-f017:**
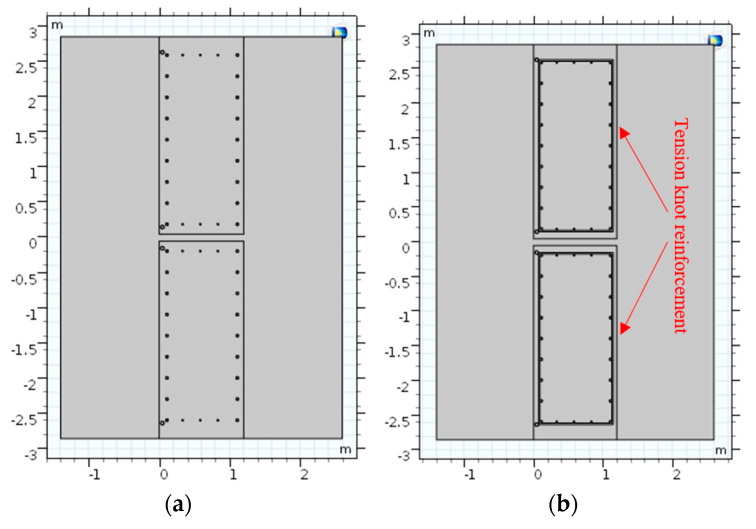
Two-dimensional profile of the model; (**a**) Sectional view of the model without tie bars; (**b**) Sectional view of the model with tie bars.

**Figure 18 materials-16-00561-f018:**
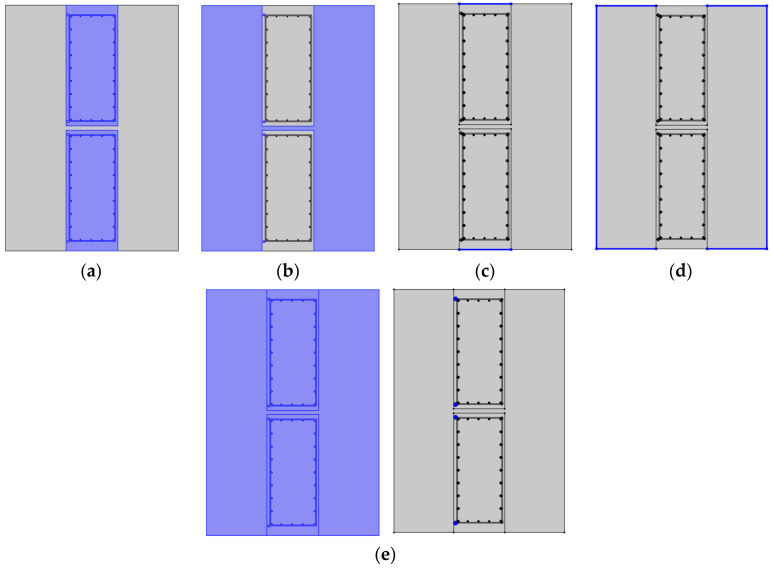
The heat transfer interface; (**a**) Wall heat transfer; (**b**) Soil heat transfer; (**c**) Insulating thermal field; (**d**) Open thermal field; (**e**) Initial temperature field.

**Figure 19 materials-16-00561-f019:**
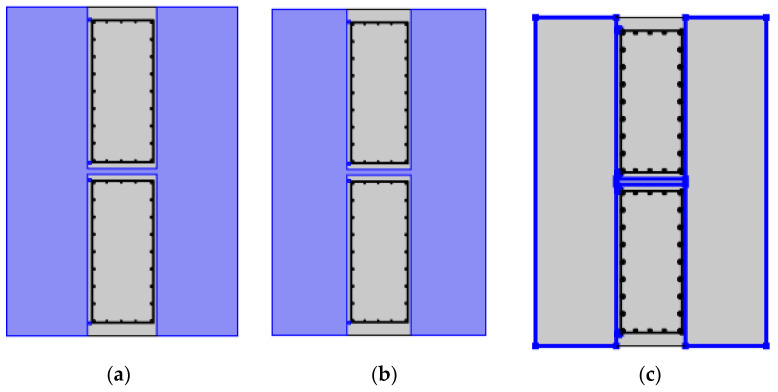
Conditions at the laminar flow interface; (**a**) Fluid properties; (**b**) Initial value; (**c**) Open boundary condition.

**Figure 20 materials-16-00561-f020:**
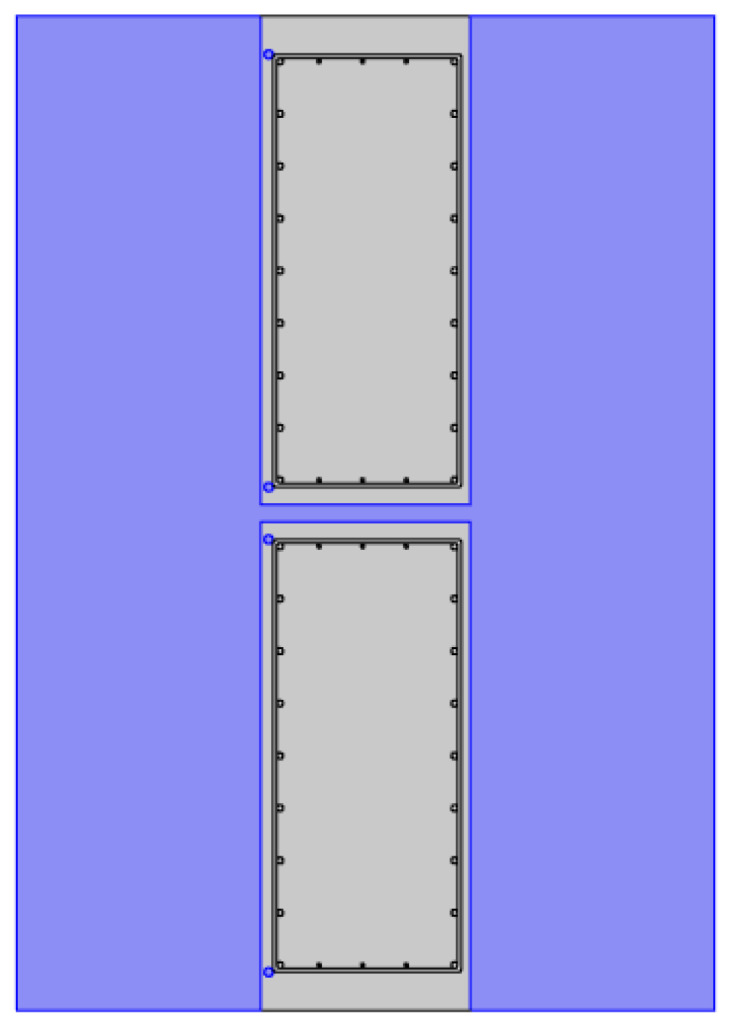
Non-isothermal flow interface.

**Figure 21 materials-16-00561-f021:**
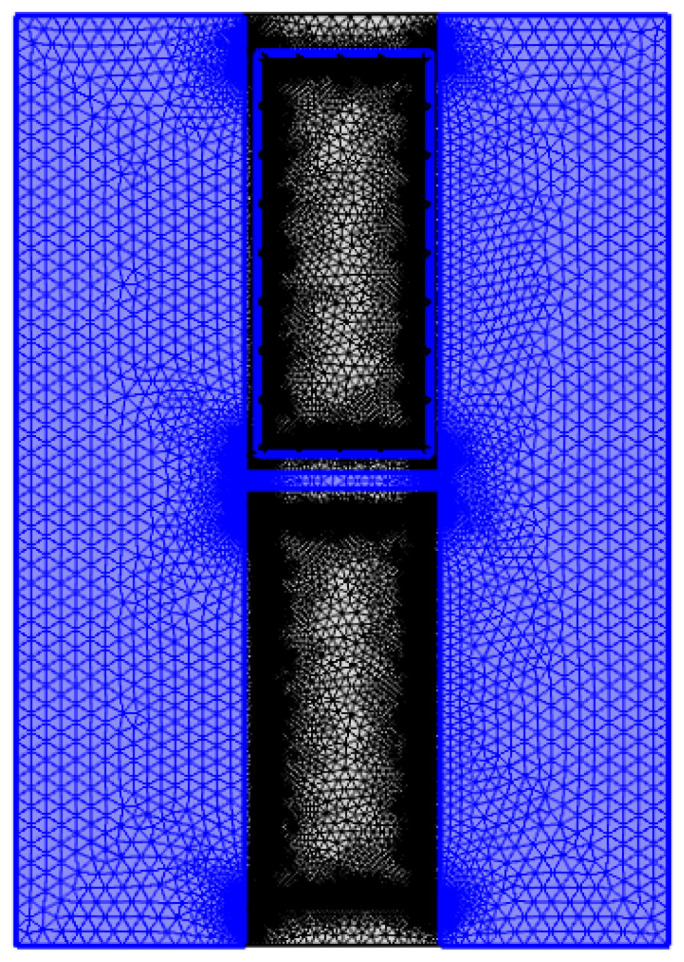
The meshing of the model.

**Figure 22 materials-16-00561-f022:**
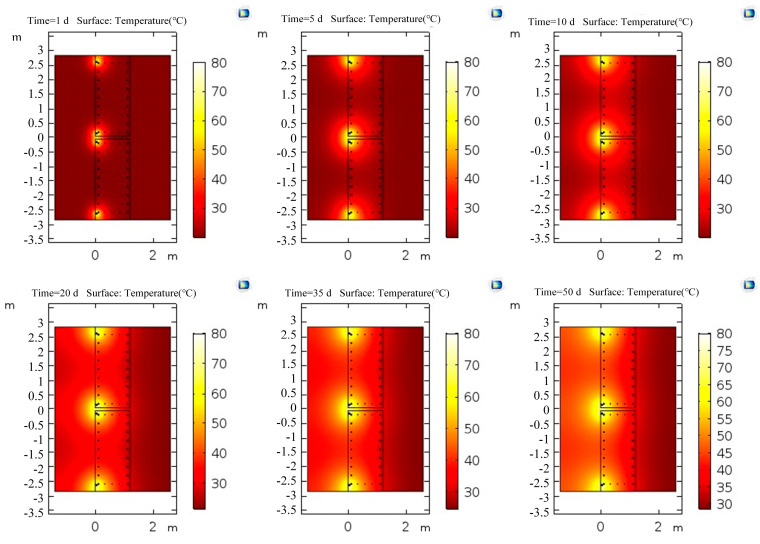
Distribution of temperature field in a model without tension reinforcement.

**Figure 23 materials-16-00561-f023:**
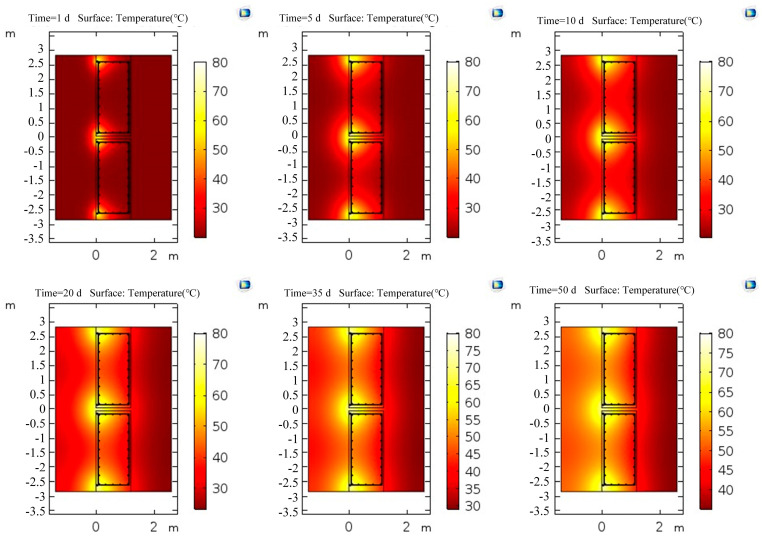
Distribution of temperature field in a model with tension reinforcement.

**Figure 24 materials-16-00561-f024:**
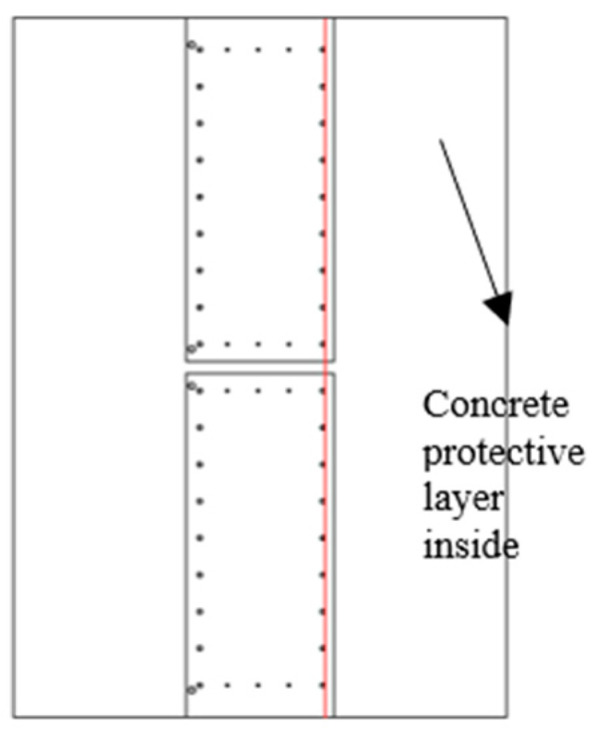
Position selection diagram of longitudinal section line.

**Figure 25 materials-16-00561-f025:**
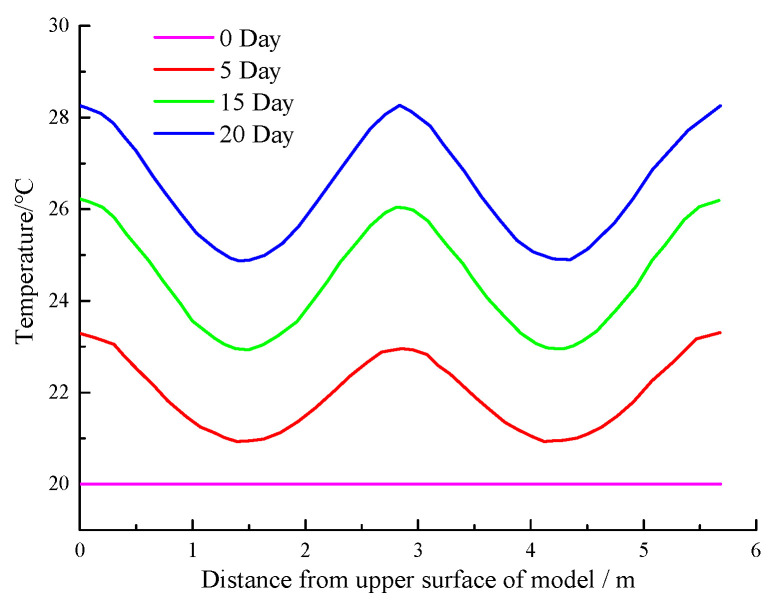
The temperature distribution curve of the longitudinal section line.

**Figure 26 materials-16-00561-f026:**
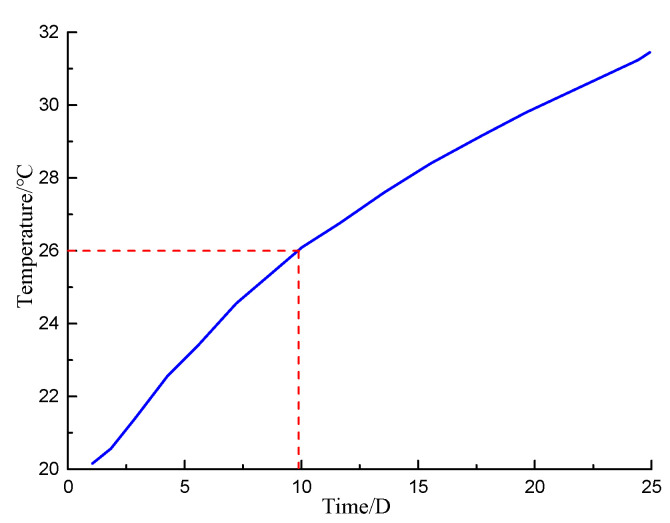
Temperature change curve of cable layout position.

**Figure 27 materials-16-00561-f027:**
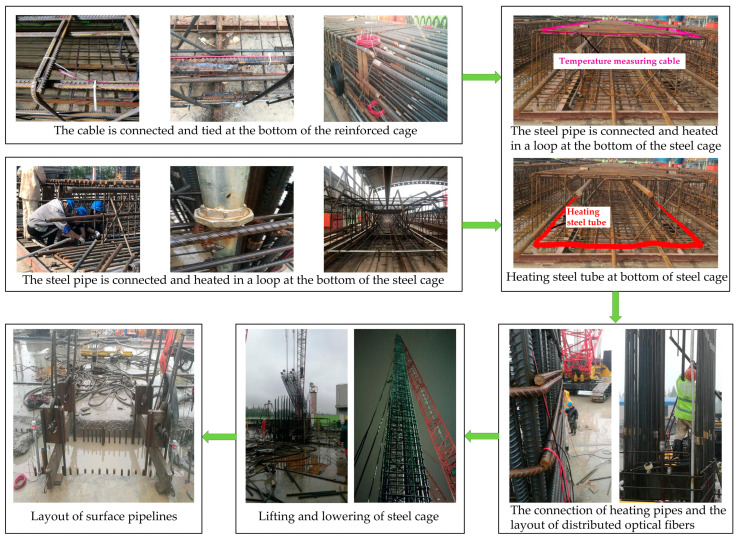
Field layout of temperature measuring optical fibers.

**Table 1 materials-16-00561-t001:** Test conditions for abnormal detection of temperature rise and fall.

Working Condition	Insulation Section	Thermal Insulation Materials	Thermal Insulation Position
A	Single segment	Insulation foam	9–10 m
B	Multi-segment	Self-adhesive insulation cotton	11–12 m; 28.5–29.5 m; 36.5–37.5 m

**Table 2 materials-16-00561-t002:** Test conditions of abnormal constant temperature detection.

Working Condition	Leakage Section	Watering Area
I-1	Single segment	14~15 m
I-2	Single segment	20~21 m
II-1	Multi-segment	17~18 m, 20~21 m
II-2	Multi-segment	17~18 m, 25~26 m

**Table 3 materials-16-00561-t003:** Pipeline layout scheme.

Test Number	Pipe Material	Connection Mode	Mode of Fixation	Segmented Docking Mode
1	Galvanized steel pipe	Tap + articulated	U-shaped steel bar welding	Stainless steel bellows
2	PPR + PE-RT	Welding + directly	Wire binding	Directly
3	Galvanized steel pipe	Tap + directly	Wire binding	Stainless steel bellows

**Table 4 materials-16-00561-t004:** Optical cable connectivity test table.

Pigtail Number	Fiber Type	Connectivity	Survival Properties
No.1 Outside the wall	Armored	Fiber is broken at 32 m	Failure
No.2 Outside the wall	Armored	Fiber is broken at 6 m	Failure
No.1 Inside the wall	Armored	Fiber is broken at 212 m	Available
No.2 Inside the wall	Armored	Fiber is broken at 212 m	Available
No.3 Inside the wall	Heating	Fiber is broken at 145 m	Available
No.4 Inside the wall	Heating	Fiber is broken at 104 m	Available

## Data Availability

Data presented in this study are available on request from the corresponding author.

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
