# Peer review of "Conceptual Model, Experiment and Numerical Simulation of Diaphragm Wall Leakage Detection Using Distributed Optical Fiber"

_materials, 2023, doi:10.3390/ma16020561_

Round 1
Reviewer 1 Report
This paper discusses a conceptual model of active and passive thermal lead detection methods.
1. There are some grammar mistakes throughout the manuscript.
2. What is the composition of the steel tubes? Where were the steel tubes fabricated?
3. Please give more detail about the temperature monitoring system. For instance, what company produced the device?
4. What composition is the measuring optical cable?
5. Please give more detail about the infrared thermal imager.
6. I do not see how Figures 2 and 3 add to the manuscript.
7. As for Figure 5, a schematic would be much better suited than the currently shown images. I honestly do not understand their significance to the text.
8. For Figure 6, could the authors add a colored temperature scale bar alongside the thermal image?
9. Figure 8 provides no context. Where are the axes values? How do the authors know that this is anomaly? Is the data point outside of a standard deviation of the mean of the data?
10. Please write the text in Figure 9 in English, or at least provide an English translation in the figure caption.
11. I do not see the significance of Figure 23.
12. Please replace Figures 24-26 with schematic diagrams.
13. Please provide equations used for the models employed in Figure 19.
14. Upon reading the Conclusions section, I am left with asking "so what?". Please better explain how this work resulted in an advance in the field.
Unfortunately, this article has some serious flaws and I must therefore reject it as is.
Reviewer 2 Report
Accepted
Author Response
Thank you very much.
Reviewer 3 Report
The authors examined the active and passive thermal leak detection methods by both experimentally and numerically. Experiments are performed using a heating system and an optical fiber temperature measurement system to verify the thermal leakage detection systems. Numerical simulations were performed to understand the mechanism of the detecting method.
Comments:
1. 1) Need more detailed schematic diagram of the physical model of the problem. Figure 1 is not enough to represent the model.
2) Give the clear motivation of the present study.
3) The literature review is inadequate. The authors must provide detailed survey of the literature of the present problem.
4) Figure.9 is not clearly visible. Provide a clear figure.
5) The mathematical model (partial differential Equations) is not clearly described in the section 4.1.1. Introduction to COMSOL Software and Conjugate Heat Transfer Module. The authors should derive the equations and conditions here.
6) What kind of numerical technique is used for the model equations? Give the details.
7) What is the convergence criteria for the obtained solution.
8) What are the numerical schmemes are used for conjugate heat transfer module?
9) What are the condition at three interfaces? How do the calculation carry out?
10) Are the conditions same for three interface (Heat transfer interface, laminar flow interface, and nonisothermal flow interface)?
11) Conclusion should be concise and brief.
12) Check the language of the paper.
Round 2
Reviewer 1 Report
The authors have put forth a good effort to address my comments and now the paper is suitable for publication. I accept it as is.
Reviewer 3 Report
The authors revised paper well. It is now suitable for publication.